# A Negative Capacitance Field-Effect Transistor with High Rectification Efficiency for Weak-Energy 2.45 GHz Microwave Wireless Transmission

**DOI:** 10.3390/mi16010058

**Published:** 2024-12-31

**Authors:** Hualian Tang, Ailan Tang, Weifeng Liu, Jingxiang Huang, Jianjun Song, Wenjie Sun

**Affiliations:** 1State Key Laboratory of Wide-Bandgap Semiconductor Devices and lntegrated Technology, School of Microelectronics, Xidian University, Xi’an 710071, China; hltang@xidian.edu.cn (H.T.); wfliu@mail.xidian.edu.cn (W.L.); jianjun_79_81@xidian.edu.cn (J.S.); 21111212964@stu.xidian.edu.cn (W.S.); 2School of Electronic Science and Technology, Xidian University, Xi’an 710071, China; hjx986234675@gmail.com

**Keywords:** 2.45 G, wireless energy transfer, weak energy density, NCFET, rectification efficiency, sentaurus TCAD

## Abstract

This paper proposes and designs a silicon-based negative capacitance field effect transistor (NCFET) to replace conventional MOSFETs as the rectifying device in RF-DC circuits, aiming to enhance the rectification efficiency under low-power density conditions. By combining theoretical analysis with device simulations, the impacts of the ferroelectric material anisotropy, ferroelectric layer thickness, and active region doping concentration on the device performance were systematically optimized. The proposed NCFET structure is tailored for microwave wireless power transmission applications. Based on the optimized NCFET, a half-wave rectifier circuit employing a novel diode connection configuration was constructed and verified through transient simulations. The results show that at a microwave frequency of 2.45 GHz, the designed NCFET rectifier achieves rectification efficiencies of 16.1% and 29.75% at input power densities of −10 dBm and −6 dBm, respectively, which are 7.15 and 2.3 times higher than those of conventional silicon-based MOS devices. Furthermore, it significantly outperforms CMOS rectifiers reported in the literature. This study demonstrates the superior rectification performance of the proposed NCFET under low-power density conditions, offering an efficient device solution for microwave wireless power transmission systems.

## 1. Introduction

With the widespread application of Wireless Sensor Networks (WSNs) and Radio Frequency Identification (RFID) systems, the challenge of providing a reliable power supply to low-power circuit devices has become increasingly prominent. Traditional power supply methods, such as batteries and wired connections, face significant limitations. These include high costs, frequent maintenance requirements, and an unsuitability for specific scenarios where the access to physical power sources is restricted or impractical [1,2,3]. To address these challenges, Microwave Wireless Power Transfer (MWPT) has emerged as an innovative energy transfer technology, attracting considerable attention from both academia and industry. This technology eliminates the spatial constraints of traditional power transmission methods, offering broad applications in smart devices, remote monitoring, and the Internet of Things (IoT) [4,5,6].

In recent years, significant progress has been made in MWPT technology, with numerous studies focusing on optimizing the performance of receiving antennas [7,8,9] and the topology of peripheral rectifier circuits [10,11,12]. These studies have improved the rectification efficiency by enhancing the antenna gain, impedance matching networks, and the peripheral structure design of rectifier circuits. However, compared to the optimization of peripheral circuits, research on the core rectifier components themselves remains relatively limited. Most newly developed rectifier devices are aimed at medium-to-high-power density environments, with a primary focus on improving the rectification efficiency under higher input power densities. For example, Joseph et al. [13] proposed a novel wire-bonded GaN rectifier suitable for high-power wireless power transmission. This microwave rectifier consists of high-breakdown voltage GaN rectifier devices designed for high-power operation and a novel impedance matching technique, achieving a maximum efficiency of 61.2% at 39 dBm, demonstrating excellent performance in the high-power range. Similarly, Zhai et al. [14], based on the diode rectification principle, developed a GeSnOI folded space charge region Schottky diode for 2.45 GHz microwave wireless power transmission, achieving a peak rectification efficiency of 77.2% at high power input levels above 20 dBm. Chek et al. [15] proposed using GaN-coated cathodes to replace the existing BaO coating for modifying high-frequency low-power magnetron cathodes, ultimately achieving an efficiency of 88.37% at 37 dBm, representing a 36.57% improvement compared to the original coating. Wang et al. [16] introduced GaAs Schottky barrier diodes with a lower series resistance and higher breakdown voltage, achieving a rectification efficiency of 91% at 37 dBm. In contrast, studies targeting the optimization of the rectifier device performance in low-power density environments are relatively rare. The lack of breakthroughs in performance improvements under weak energy density conditions has become a significant bottleneck, limiting the widespread application of MWPT systems in energy-constrained scenarios.

Under low-power density conditions, traditional rectifier devices such as Schottky diodes and MOSFETs often struggle to operate efficiently. Although Schottky diodes perform well under certain conditions, their rectification efficiency is relatively low under weak energy densities, and they are incompatible with CMOS processes [17]. Similarly, traditional MOSFETs are significantly limited in performance under a low input signal energy, leading to a generally low rectification efficiency. These limitations hinder the application of MWPT systems in energy-scarce scenarios. Therefore, designing and optimizing rectifier devices for weak energy environments has become a critical research direction for improving the rectification efficiency and overcoming technical bottlenecks, while also presenting new opportunities for the practical deployment of MWPT technology.

Among potential solutions, negative capacitance field effect transistors (NCFETs) offer promising advantages. Compared to traditional MOSFETs, NCFETs exhibit unique properties, including an extremely low sub-threshold swing (SS), high switching current ratios, simple device structures, and full compatibility with standard silicon semiconductor processes. These characteristics position NCFETs as strong candidates for improving the rectification efficiency in low-power density applications [18].

This paper focuses on addressing the rectification challenges under weak energy density conditions at the 2.45 GHz microwave frequency band. A silicon-based N-type NCFET was proposed and designed as the core rectifier device to optimize RF-DC rectifier circuits. An RF-DC half-wave rectifier circuit was built for transient analysis, comparing the rectification performance of NCFETs against conventional silicon-based MOSFETs in low-energy density ranges. This study also evaluated the NCFET’s performance relative to several CMOS devices reported in the literature. The results demonstrate the superior rectification efficiency of NCFETs, highlighting their potential to enhance MWPT systems and broaden their application scope in weak energy environments.

## 2. Si-Based NCFET Design

### 2.1. Basic Working Principle of NCFET

NCFETs retain the thermal excitation process of conventional FETs, with modifications made to the gate structure by introducing metal/ferroelectric/metal (MFM) or metal/ferroelectric (MF) layers to the gate oxide. Figure 1 shows the resulting MFMIS and MFIS structures.

The MFMIS structure, due to an internal floating metal gate, achieves a more uniform voltage amplification along the channel [19]. This study modeled the MFMIS-based NCFET using the Landau–Khalatnikov (L-K) equation as follows, commonly applied in theoretical studies and simulation [20,21].

(1)
ρdP/dt+∇ρU=0


Here, 
ρ
 is the dissipation resistance, dependent on the material and bias voltage. Combining Equations (1) and (2), the electric field–polarization relationship can be derived.

(2)
$$E=2\alpha P+4\beta P^{3}+6\gamma P^{5}+\rho dP/dt$$


The relationship between the voltage 
$V_{FE}$
 across the ferroelectric body and the thickness of the ferroelectric layer 
$T_{FE}$
 can be further derived from Equation (Equation 2) as follows.

(3)
$$V_{FE}=\left(2\alpha Q+4\beta Q^{3}+6\gamma Q^{5}+\rho \partial P/\partial t\right)\times T_{FE}$$


When the electric field is small, the charge on the capacitor is 
$Q=P+\epsilon_{0}E\approx P$
. On this basis, the expression for the ferroelectric capacitance can be obtained below.

(4)
$${C_{FE}}^{-1}=2\alpha T_{FE}+12\beta T_{FE}P^{2}+30\gamma T_{FE}P^{4}$$


Neglecting the higher order terms, 
$C_{FE}$
 can be transformed into the below.

(5)
$$C_{FE}=dQ/dV_{FE}\approx 1/\left(2\alpha \times T_{FE}\right)$$


This indicates that 
$C_{FE}$
 is primarily influenced by the ferroelectric layer thickness and the material’s anisotropy parameter 
α
. The equivalent capacitance model of the NCFET is shown in Figure 2.

The relationship between voltages can be obtained using the capacitance divider relationship below. It can be seen that when 
$C_{FE}$
 is stabilized in the negative capacitance region (
$V_{FE}$
 < 0), at which time 
$V_{GMOS}$
 > 
$V_{GNC}$
, relative to the intrinsic gate voltage, 
$C_{FE}$
 can play a role in gate voltage amplification [22].

(6)
$$V_{GMOS}=V_{GNC}-V_{FE}$$


In addition, the gate capacitance 
$C_{g}$
 of the NCFET consists of 
$C_{FE}$
 and the insulating layer capacitor 
$C_{ox}$
 in series; therefore, we have the equation below. When 
$C_{FE}$
 is negative, 
$C_{g}$
 increases, enhancing the channel charge concentration and resulting in a steeper sub-threshold swing and higher current gain.

(7)
$$1/C_{g}=1/C_{FE}+1/C_{ox}$$


### 2.2. Simulation of Device Structure Design and Parameter Optimization

Studies have shown that a stable negative capacitance can be achieved by matching the appropriate positive capacitance, and the degree of capacitance matching will directly affect the electrical performance of NCFETs [23]. The NCFET capacitance matching situation is shown in Figure 3. When the capacitance is mismatched, the NCFET exhibits a locally steep sub-threshold swing and there is significant hysteresis; when the capacitance is matched, the NCFET negative capacitance effect is stable and there is no current hysteresis phenomenon, and the higher the degree of capacitance matching, the better the electrical performance of the device, with a lower sub-threshold swing and higher current gain. Therefore, the key to designing NCFETs is capacitance matching and improving the degree of capacitance matching as much as possible.

The Si-based NCFET structure designed in this paper is shown in Figure 4, and the important simulation parameters are marked in the figure. In order to avoid the influence of a short channel effect as far as possible, the channel length was set to 0.2 um; considering the difficulty in the actual process of preparation, the thickness of the gate oxide layer was set to 2 nm; and regarding the choice of the gate oxide layer material, this paper chose 
$HfO_{2}$
, which has a higher dielectric constant compared to 
$SiO_{2}$
, as the gate oxygen layer, as it has a higher 
$C_{ox}$
 at the same thickness and can be used at a lower ferroelectric layer thickness. Good capacitance matching can be achieved with a lower ferroelectric layer thickness. Since 
$HfO_{2}$
- and 
$ZrO_{2}$
-based ferroelectric materials are highly CMOS process-compatible and scalable, and can also exhibit good ferroelectricity when the ferroelectric layer is thin, 
$Hf_{0.5}Zr_{0.5}O_{2}$
 (HZO) was chosen as the gate ferroelectric material in this paper, and its anisotropy parameters, 
α
, 
β
, and 
γ
, are listed in Table 1 [24,25].

According to the analysis in Section 2.1, under the condition that the ferroelectric material has been selected, i.e., a fixed ferroelectric parameter 
α
, the ferroelectric capacitance can be controlled by adjusting the thickness of the ferroelectric layer to achieve a better capacitance matching effect and improve the electrical performance of the NCFET. Using sentaurus TCAD software (vO-2018.06) in accordance with the ferroelectric material parameters given above, the Si-based NCFET simulation model and mesh division are shown in Figure 5. A denser mesh was set in the key areas of interest, such as the channel and gate oxygen layer, and a coarser mesh was set at the substrate to increase the simulation speed while ensuring the simulation results are as accurate as possible [26]. Based on this structure, the simulation analyzed the transfer characteristics, threshold voltage, gate capacitance, gate voltage amplification factor, switching current ratio, and trans-conductance of NCFETs with different ferroelectric layer thicknesses to obtain the optimal device structure parameters for the comprehensive selection of performance indicators.

Figure 6 shows the transfer characteristic curves of NCFETs with different ferroelectric layer thicknesses, where the left-hand side is the logarithmic coordinate system and the right-hand side is the ordinary coordinate system with an applied drain source voltage of 0.1 V. From the ordinary coordinate system, it can be seen that the open-state current gradually increases as the ferroelectric layer thickness increases, and from the logarithmic coordinate system, it is obvious that the sub-threshold swing, ss, of NCFETs is significantly lower than that of conventional MOS transistors. With a lower sub-threshold swing, the NCFET has a higher switching current ratio compared to conventional MOSFETs for a similar off-state current.

Figure 7 shows the variation in the gate capacitance, Cg, with the gate voltage for NCFET devices with different ferroelectric layer thicknesses. From the figure, it can be seen that the gate capacitance of the device increases with the same gate voltage after adding ferroelectric materials to the gate section, and the thicker the ferroelectric layer, the more obvious the effect of increasing the gate capacitance, Cg, and when the ferroelectric layer thickness is 8 nm, there is an obvious “capacitance spike”, which is a typical feature of NCFETs and also explains why NCFETs can break the thermodynamic limit of sub-threshold swing of 60 mv/dec.

Figure 8 shows the variation in the trans-conductance, Gm, with the gate voltage for NCFET devices with different ferroelectric layer thicknesses. From the figure, it can be seen that NCFETs have higher Gm values at lower gate voltages compared to conventional MOS transistors, and as the thickness of the ferroelectric layer increases, the Gm improvement effect becomes more pronounced, indicating that NCFET devices have a higher current gain at low gate voltages, allowing the device current to increase rapidly and the channel to open faster.

Ferroelectric negative capacitance transistors can produce a “gate voltage amplification effect” due to the introduction of ferroelectric materials in the gate section, specifically when the change in potential at the channel surface of the transistor is greater than the change in the gate voltage (
$A_{V}=\left(d\phi_{s}\right)/\left(dV_{G}\right)>1$
). Figure 9 shows the comparison of the gate voltage amplification factor Av for different ferroelectric layer thicknesses. It can be seen that the gate amplification coefficient Av of the conventional MOS transistor (
$T_{FE}$
 = 0 nm) is less than one throughout the gate voltage range, while after adding the ferroelectric layer, the transistor shows a gate amplification effect (Av > 1) in part of the voltage range, and the gate amplification effect becomes more obvious as the thickness of the ferroelectric layer gradually increases, and when the thickness of the ferroelectric layer is 8 nm, the transistor has a gate amplification effect (Av > 1) in the voltage range of 0–0.5 v, and the maximum gate amplification factor can reach 1.81.

In summary, the NCFET with a ferroelectric layer thickness of 8nm was finally chosen as the device for the subsequent rectification characteristics simulation. A comparison of its output characteristic curve and that of a conventional MOSFET under the same conditions is given in Figure 10. As can be seen from the figure, the NCFET has a higher output current than the conventional MOSFET with the same applied gate voltage, and the lower the gate voltage, the more pronounced the output current increase, with a nearly twofold increase in the output current at an applied gate voltage of 0.25 V.

## 3. Rectifier Circuit Design and Simulation

### 3.1. Rectifier Circuit Construction and Design of New Diode Connection Method

In order to analyze the rectification efficiency of the Si-based NCFET device designed in this paper when applied to 2.45 GHz microwave wireless energy transmission, and to consider the complexity of the simulated circuit, a half-wave rectifier circuit topology was built, as shown in Figure 11, with the rectifier devices connected to the circuit in the form of diodes. An AC voltage source with an internal resistance of 50
ω
 was used to simulate the microwave energy signal received by the antenna as the input source of the rectifier circuit, and a resistor, R1, was used as the output load, with a capacitor, C1, acting as a voltage regulator [27].

When the rectifier is connected to the circuit, instead of the traditional diode connection where the gate and drain are shorted as the input and the source and liner are shorted as the output, we propose a new connection method; the gate, drain, and liner are connected as the equivalent input and the source is the equivalent output (Figure 12). The bias voltage 
$V_{SB}$
 between source lines is not zero with the new diode connection; thus, there is a substrate bias effect [28], calculated as shown below.

(8)
$$V_{TH}=V_{TH0}+\gamma \left(\sqrt{\left|2\phi_{F}+V_{SB}\right|}-\sqrt{\left|2\phi_{F}\right|}\right)$$


The application of a forward voltage results in a negative 
$V_{SB}$
, which reduces the forward turn-on voltage and increases the forward turn-on current. Conversely, when a reverse voltage is applied, a positive 
$V_{SB}$
 is generated, which raises the channel turn-on voltage, suppressing the reverse leakage current and thereby enhancing the rectification efficiency. The DC simulation results for the two connection methods are presented in Figure 12. The current trends shown in the figure indicate that the new diode connection enhances the forward conduction current while effectively suppressing the reverse leakage current.

### 3.2. Transient Simulation and Rectification Efficiency Analysis

The rectification efficiency can be calculated using the input and output power over one cycle, as shown in Equation (Equation 9).

(9)
$$\eta =\left(1/T\int_{0}^{T}I_{out}\times V_{out}\right)/\left(1/T\int_{0}^{T}I_{in}\times V_{in}\right)\times 100\%$$


A transient simulation of the circuit was performed using the Sentaurus TCAD tool. The waveform at 18.5 ns–19.1 ns of stabilization is shown in Figure 13a, from which it can be seen that the input AC signal, after the rectification circuit, was converted into a DC pulsating signal, achieving a good rectification effect. The peak-to-peak voltage value was 0.2 v, the load resistance was 50 k
Ω
, and the load capacitance was 0.5 pF. The local magnification of the input and output voltages in a single cycle is shown in Figure 13b, where 
δ
V is the output pulsating voltage variation; a smaller 
δ
V indicates that the DC signal at the output is more stable, i.e., a better rectification effect. Figure 13c shows the comparison of the input and output power after the integration calculation according to Equation (Equation 9). The rectification efficiency can be obtained by integrating the output and input power over the same time period and then comparing them.

Figure 14 shows the rectification efficiency curve of NCFET devices calculated according to the above method, while a conventional MOS transistor was selected as a control group under the same conditions, and the rectification efficiency of CMOS devices at weak energy densities in several papers [29,30,31,32,33,34] are marked in the figure, and the specific parameters are shown in Table 2.
micromachines-16-00058-t002_Table 2Table 2CMOS rectification performance summary table.ReferenceTechniqueRectifier TopologyFrequency (GHz)Rectification Efficiency (% at Load @ Input Energy Density)This work200 nm NMOS1-stage half-wave2.4516.1% for 20 K
Ω
 @−10 dBm[29]90 nm CMOS4-stage Dickson2.451% for 1 M
Ω
 @−8.06 dBm[30]300 nm CMOS3-stage Dickson0.951.5% for 200 K
Ω
 @−14 dBm[31]180 nm CMOS5-stage on-chip inductor0.914.46% for 200 K
Ω
 @−9 dBm[32]350 nm CMOS1-stage full-wave0.9515.4% for 1 M
Ω
 @−9 dBm[33]65 nm CMOS3-stage Dickson5213% for 10 M
Ω
 @−10 dBm[34]180 nm CMOS5-stage Dickson2.637% for 500 K
Ω
 @−15.4 dBm

As can be seen from the figure, the NCFET device designed in this paper widened the rectification range at a weak energy density by about 10dBm compared to the conventional MOS transistor, while the rectification efficiency in the weak energy range was also improved by different magnitudes. An input signal power density −10 dBm rectification efficiency of up to 16.1%, found under the same conditions using the MOSFET 7.15 times, compared with other CMOS rectifier devices reported in the literature, also reflects a more excellent rectification performance under a weak energy density, while in the high-energy density range of 10 dbm or more, the NCFET rectification efficiency was also comparable to the MOS. As can be seen, the NCFET compensates for the lower rectification efficiency of conventional MOS transistors when operating at weak energy densities.

## 4. Conclusions

In this paper, a Si-based NCFET was proposed and designed to replace the conventional MOS transistor as a rectifier device in RF-DC circuits. This paper presents the basic working principle of an NCFET and establishes its threshold voltage model. The effects of the ferroelectric material anisotropy, ferroelectric layer thickness, and doping concentration in the active region on the device performance were analyzed. Based on electrical simulations, the device structure was optimized for improved performance in rectifier circuits under low-energy density conditions. The simulation results show that the NCFET has a wider rectification range and excellent rectification performance in the weak energy density range, with rectification efficiencies of 16.1% and 29.75% at input power densities of −10 dBm and −6 dBm, respectively, which are 7.15 times and 2.3 times higher than those of an Si-based MOS under the same conditions and also reflect the rectification efficiencies of CMOS devices when compared to those reported in several papers. The rectification efficiency of the designed NCFETs at the weak energy density of 2.45 G also reflects the rectification advantage of the model discussed in this paper.

## Figures and Tables

**Figure 1 micromachines-16-00058-f001:**
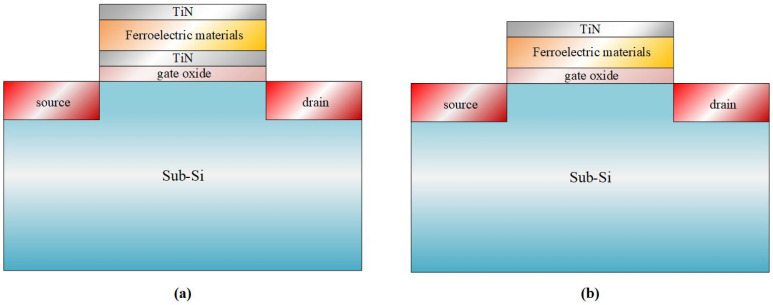
Schematic diagram of the NCFET for (**a**) MFMIS and (**b**) MFIS structures.

**Figure 2 micromachines-16-00058-f002:**
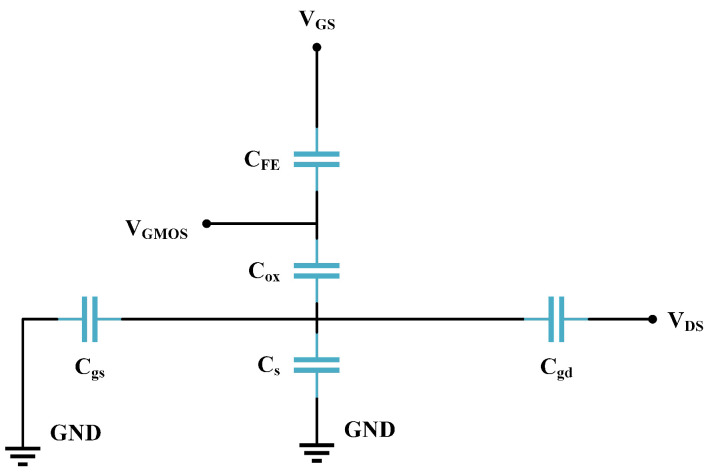
NCFET equivalent capacitance model.

**Figure 3 micromachines-16-00058-f003:**
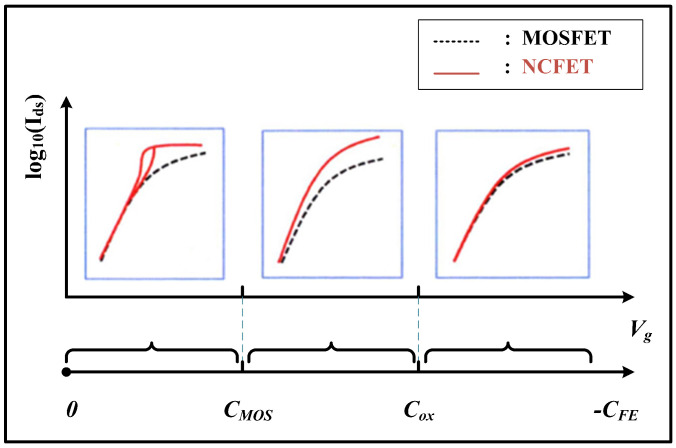
Capacitance matching of NCFETs [23].

**Figure 4 micromachines-16-00058-f004:**
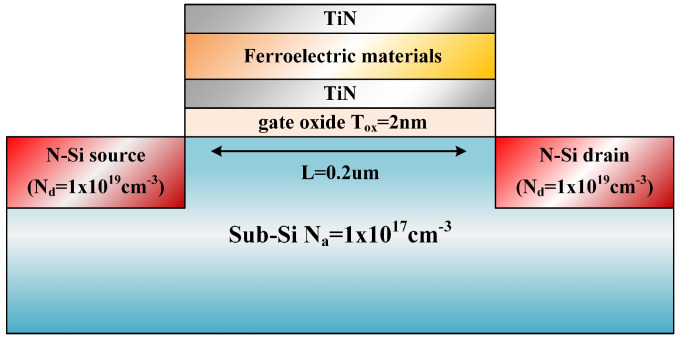
NCFET device structure and simulation’s key parameter values.

**Figure 5 micromachines-16-00058-f005:**
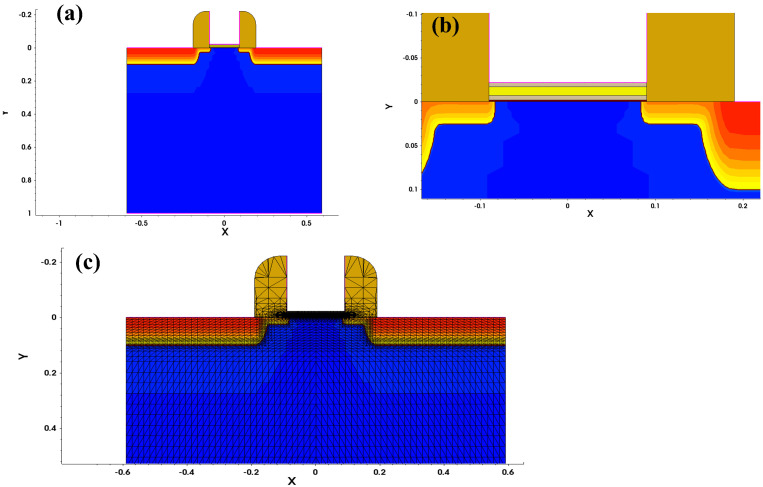
Sentaurus TCAD simulation model of NCFET. (**a**) Overall structure of the NCFET device. (**b**) Enlarged view of the NCFET gate region. (**c**) Simulation mesh diagram of the NCFET.

**Figure 6 micromachines-16-00058-f006:**
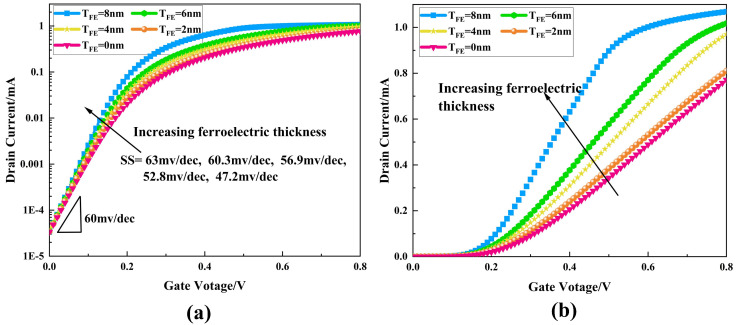
Transfer characteristic curves of NCFETs with different ferroelectric layer thicknesses (**a**) with logarithmic coordinates and (**b**) with linear coordinates.

**Figure 7 micromachines-16-00058-f007:**
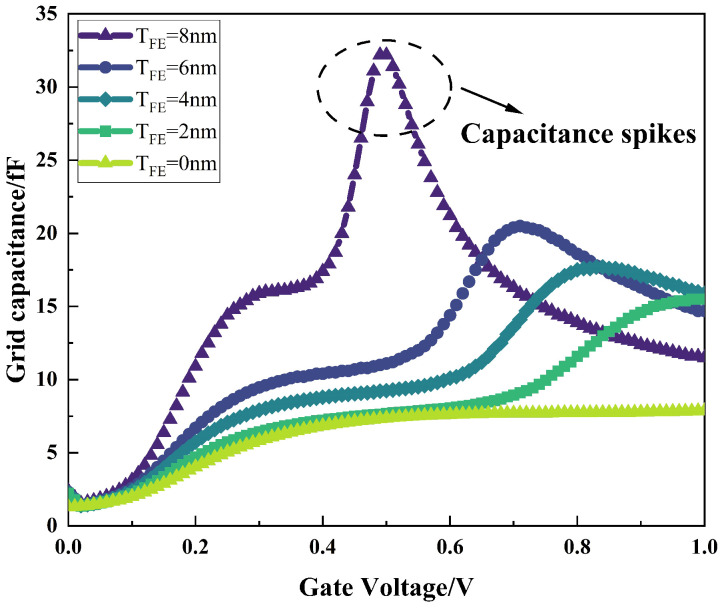
Variation in gate capacitance, Cg, with gate voltage for different ferroelectric layer thicknesses.

**Figure 8 micromachines-16-00058-f008:**
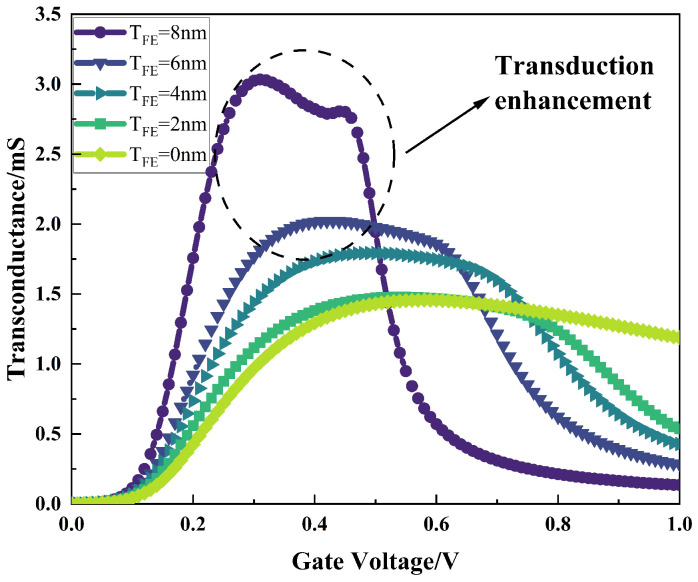
Variation in trans-conductance, Gm, with gate voltage for different ferroelectric layer thicknesses.

**Figure 9 micromachines-16-00058-f009:**
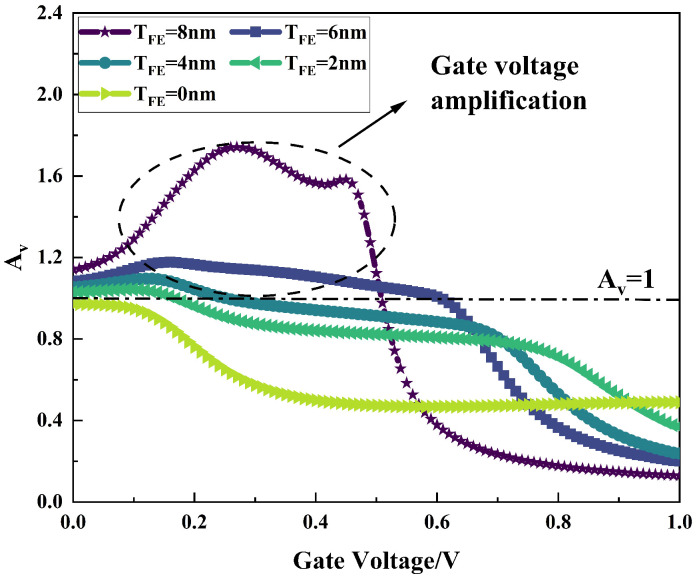
Variation in gate amplification factor Av with gate voltage for different ferroelectric layer thicknesses.

**Figure 10 micromachines-16-00058-f010:**
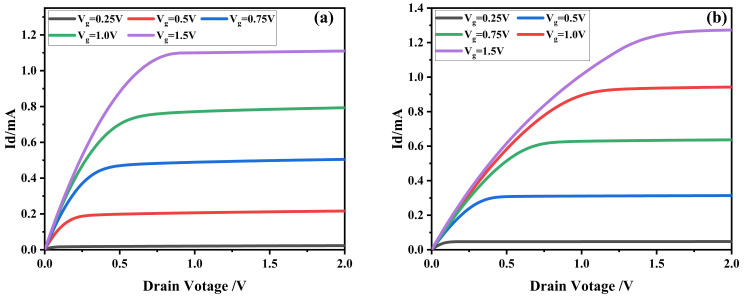
Comparison of MOSFET and NCFET output characteristic curves. (**a**) MOSFET output characteristics. (**b**) NCFET output characteristics.

**Figure 11 micromachines-16-00058-f011:**
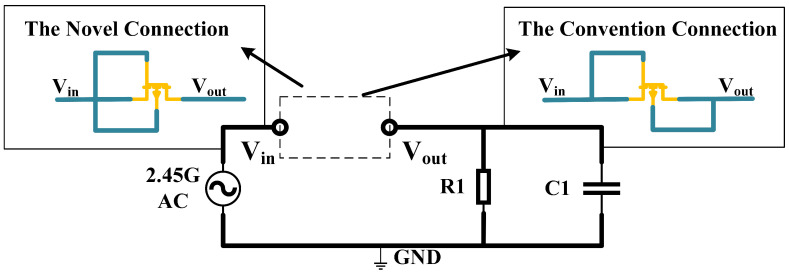
Schematic diagram of a half-wave rectifier circuit.

**Figure 12 micromachines-16-00058-f012:**
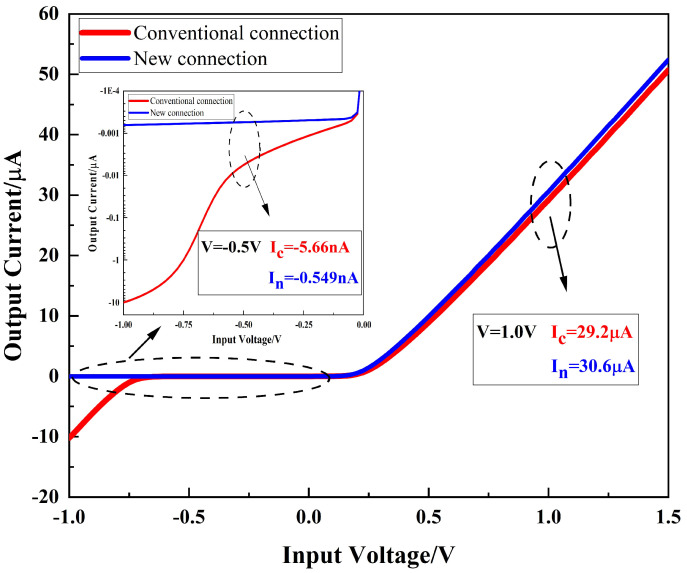
I-V diagram for DC simulation with two diode connections.

**Figure 13 micromachines-16-00058-f013:**
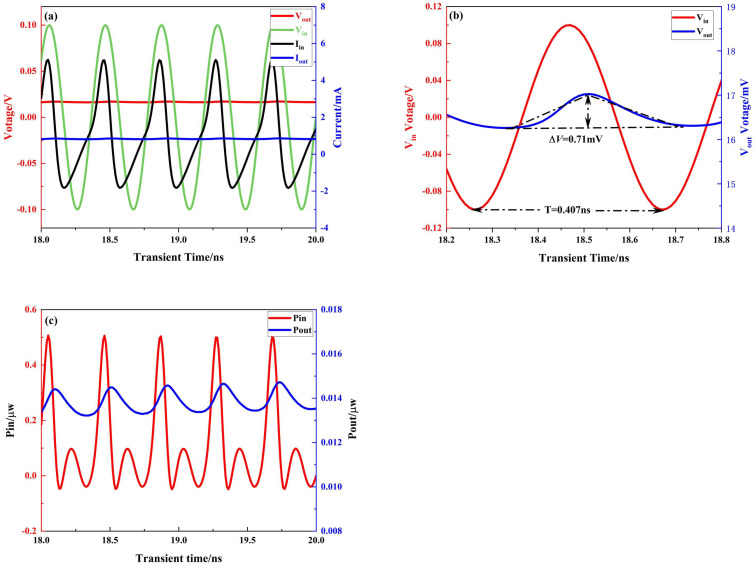
(**a**) Transient simulation input and output voltage and current waveforms. (**b**) Transient simulation single-cycle input and output voltage local magnification. (**c**) Transient simulation input and output power diagram.

**Figure 14 micromachines-16-00058-f014:**
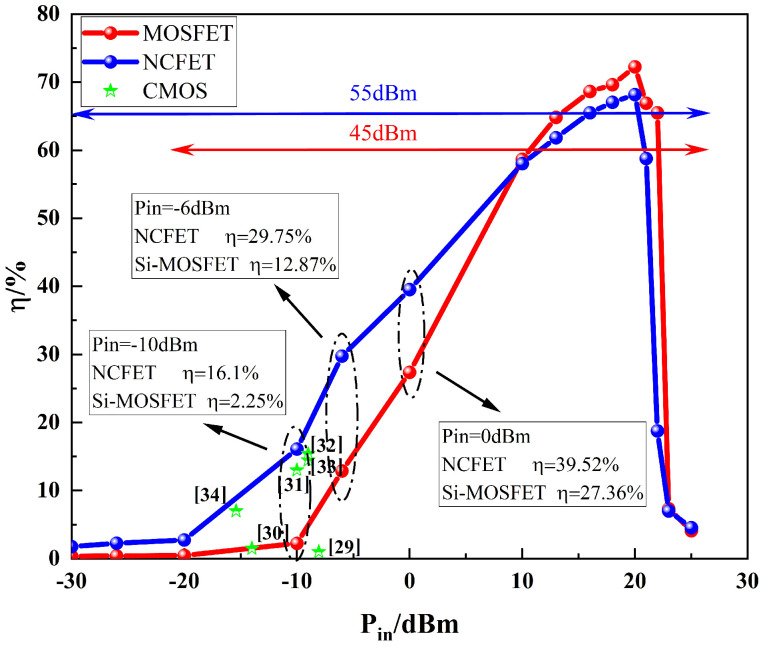
A comparison of the NCFET with the conventional MOS and CMOS rectification efficiency as reported in the literature.

**Table 1 micromachines-16-00058-t001:** Anisotropic parameters of 
$Hf_{0.5}Zr_{0.5}O_{2}$
 (HZO).

Anisotropy Parameters	Value	Unit
α	$-1.1\times 10^{11}$	Cm/F
β	$2.5\times 10^{11}$	Cm^5^/FC^2^
γ	0	Cm^9^/FC^4^
g	$1\times 10^{-4}$	Cm^3^/F

## Data Availability

The original contributions presented in this study are included in the article. Further inquiries can be directed to the corresponding author.

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
