# Peer review of "A Negative Capacitance Field-Effect Transistor with High Rectification Efficiency for Weak-Energy 2.45 GHz Microwave Wireless Transmission"

_micromachines, 2024, doi:10.3390/mi16010058_

Round 1
Reviewer 1 Report
Comments and Suggestions for Authors
This paper proposes a silicon-based NCFET to replace conventional MOSFET as the rectifying device in RF-DC circuits. Simulation results show that at a microwave frequency of 2.45 GHz, the designed NCFET rectifier achieves rectification efficiencies of 16.1% and 29.75% at input power densities of -10 dBm and -6 dBm,respectively, dmonstrating the superior rectification performance of the proposed NCFET under low-power density conditions.However, my concerns are:
(1) Equation (8) is not established by you, but a common channel threshold voltage model. Therefore, it is suggested to remove the corresponding sentence in Abstract.
(2) The Basic Working Principle of NCFET section seems too long, which should be shorten.
(3) The FE in VFE and TFE should be subscript, because they are subscript in text and equations.
(4) In figures, fe should be FE.
(5) The words in Figures 5 and Figure 6 are too small.
(6) The authors are encouraged to provide the device model used in TCAD to support that the simulation results are reasonable.
(7) There are some blanks at the end ot pages.
Author Response
Thank you very much for taking the time to review this manuscript. Please find the detailed responses below and the corrections highlighted in the re-submitted files.
Comments 1: Equation (8) is not established by you, but a common channel threshold voltage model. Therefore, it is suggested to remove the corresponding sentence in Abstract.
Response 1: We are sorry for the inaccurate representations. Thank you for point it out. The corresponding sentence in abstract is deleted in page 1, line 3-4.
“By combining theoretical analysis with device simulations, a threshold voltage model for NCFET was established, the impacts of ferroelectric material anisotropy, ferroelectric layer thickness, and active region doping concentration on device performance were systematically optimized.”
Comments 2: The Basic Working Principle of NCFET section seems too long, which should be shorten.
Response 2: We have re-written this part according to the Reviewer’s suggestion. A more concise discourse is used to explain principle, and unnecessary derivations are removed, which will not influence the context. And here we did not list the changes but marked in red in the revised paper, which can be found in page 2-4.
Comments 3: The FE in VFE and TFE should be subscript, because they are subscript in text and equations.
Response 3: We are sorry for our careless mistakes. Thank you for your reminder. In our resubmitted manuscript, the subscript is corrected and highlighted in blue ,which could be found in page 3-5.
Comments 4: In figures, fe should be FE.
Response 4: Thank you for your careful check, Figure 3, 6-9 is revised to make the word harmonized within the whole manuscript, which is highlighted in green in the revised paper.
Comments 5: The words in Figures 5 and Figure 6 are too small.
Response 5: Thank you for pointing it out, the word size of Figure 5 and Figure 6 is adjusted, which could be found in page 6-7.
Comments 6: The authors are encouraged to provide the device model used in TCAD to support that the simulation results are reasonable.
Response 6: Thank you for your suggestion to improve the paper. We agree that the device model used in our simulation should be provided. We have supplemented the device simulation model we used in Figure 5 with some local details, which could be found in page 6.
Comments 7: There are some blanks at the end ot pages.
Response 7: Thank you for your reminder. We found that this may be due to the fact that some of the images are too large, and we tried our best to change the size of the images to fit in. The results can be seen in the clean version without modified traces, but I feel so sorry that it seems that I can't submit more than one Word/PDF file in the system, so the submitted attachment is just the version with modified traces. Sorry again for the inconvenience.

Reviewer 2 Report
Comments and Suggestions for Authors
1- define NCFET in the title.
2- The introduction section is limited. It should be extended to include the most recent literature.
3- The format of the equations and resolution of figures should be modified. Keep space between figures and sections.
4- please compare NCFET and MOSFET at a gate voltage of 1.5V.
5- what is the cost and reliability of the NCFET compared to MOSFET?
6- How much is THE RECTIFICATION RANGE OF NCFET?
Author Response
Thank you very much for taking the time to review this manuscript. Please find the detailed responses below and the corrections highlighted in the re-submitted files.
Comments 1: define NCFET in the title.
Response 1: Thank you for your suggestion. The NCFET is now defined in the title as below.
“An NCFET (Negative Capacitance Field-Effect Transistor) with High Rectification Efficiency for Weak Energy 2.45 GHz Microwave Wireless Transmission”
Comments 2: The introduction section is limited. It should be extended to include the most recent literature.
Response 2: We sincerely appreciate your comment. We have checked the literature carefully and added more references (ref-journal 8-10) on rectifier antennas and rectifier circuits. In addition, we have added more research on rectifiers by other researchers (ref-journal 13-16), and presented their research progress and results to support our ideas. We have highlighted the supplement in page 1-2 of the revised paper.
Comments 3: The format of the equations and resolution of figures should be modified. Keep space between figures and sections.
Response 3: Thanks for your careful checks. We are sorry for our carelessness. In our re-submitted manuscript, we have adjusted the resolution of the images to more than 600 dpi according to the requirements of the journal. And the space between figures and sections is kept.
Comments 4: please compare NCFET and MOSFET at a gate voltage of 1.5V.
Response 4: Thank you for your suggestion. We have supplemented the output characteristic curves of NCFET and MOSFET under a gate voltage of 1.5V in Figure 10. And it is also highlighted in page 8 of the revised paper.
Comments 5: what is the cost and reliability of the NCFET compared to MOSFET?
Response 5: Thank you for raising this important point. As you mentioned, cost and reliability are essential considerations for industrial production. While this work primarily focuses on demonstrating the performance advantages of NCFET compared to MOSFET, we fully recognize the need to address these aspects in practical applications. Future research will focus on optimizing the fabrication process and evaluating long-term reliability to ensure the viability of NCFETs for industrial deployment.
Comments 6: How much is THE RECTIFICATION RANGE OF NCFET?
Response 6: Thank you for your comment. The rectification range of rectifier devices is also an important indicator. And our NCFET achieves a rectification range from -30 dBm to 25 dBm, totaling 55 dBm, which is nearly 20 dBm wider than that of MOSFETs under the same conditions. This improvement is primarily attributed to our device's rectification efficiency reaching 2% as early as -30 dBm in the low energy density range, whereas MOSFETs under the same conditions could not achieve the rectification efficiency above 2% until at -10 dBm. To give you a more intuitive understanding, we have provided a rectification efficiency graph with specific data below.

Round 2
Reviewer 2 Report
Comments and Suggestions for Authors
PLEASE USE THE TITLE AS FOLLOWS:
A Negative Capacitance Field-Effect Transistor with High Rectification Efficiency for Weak Energy 2.45 GHz Microwave Wireless Transmission